# Infrared Thermography Assessment of Aerobic Stability of a Total Mixed Ration: An Innovative Approach to Evaluating Dairy Cow Feed

**DOI:** 10.3390/ani13132225

**Published:** 2023-07-06

**Authors:** Burak Türkgeldi, Fisun Koç, Maximilian Lackner, Berrin Okuyucu, Ersen Okur, Valiollah Palangi, Selim Esen

**Affiliations:** 1Department of Animal Science, Tekirdağ Namik Kemal University, Tekirdağ 59030, Türkiye; 2Department of Industrial Engineering, University of Applied Sciences Technikum Wien, Hoechstaedtplatz 6, 1200 Vienna, Austria; 3Department of Biosystem Engineering, Tekirdağ Namik Kemal University, Tekirdağ 59030, Türkiye; 4Department of Animal Science, Faculty of Agriculture, Ege University, Izmir 35100, Türkiye; 5Balikesir Directorate of Provincial Agriculture and Forestry, Republic of Turkey Ministry of Agriculture and Forestry, Balikesir 10470, Türkiye

**Keywords:** total mixed ration, moisture level, infrared thermography, aerobic stability, chemical composition

## Abstract

**Simple Summary:**

An investigation of poor aerobic stability in high-moisture total mixed rations (TMRs) for livestock feed was conducted in this study. TMR quality factors were discovered, and prospective approaches to increase its fermentation properties and overall stability were investigated. Using infrared thermography (IRT) measurements, it was suggested that dairy cow feeding methods can be optimized in the field by identifying portions with a higher center temperature and maximum temperature difference values. According to the findings, there is a significant potential for IRT to be used in feed management and preservation processes in the future, resulting in higher levels of productivity for livestock.

**Abstract:**

A major objective of this study is to identify factors influencing the quality of high-moisture total mixed rations (TMRs) for livestock feed and explore possible manipulations that can enhance their fermentation characteristics and stability in order to address the problem of poor aerobic stability. Therefore, the current study utilized infrared thermography (IRT) to assess the aerobic stability of water-added TMRs in the feed bunker. By manipulating the moisture content of freshly prepared TMRs at four different levels through water addition and subjecting it to storage at two consistent temperatures, significant correlations between IRT values (center temperature (CT) and maximum temperature difference (MTD)) and key parameters such as lactic acid bacteria, water-soluble carbohydrates, and TMR pH were established. The first and second principal components together accounted for 44.3% of the variation, with the first component’s load influenced by IRT parameters, fermentation characteristics, and air exposure times, while the second component’s load was influenced by dry matter content and lactic acid concentration. The results of these studies indicate the possibility that feeding methods can be optimized by identifying portions with higher CT or MTD data using IRT measurements just before feeding dairy cows in the field. As a result, increasing the use of IRT in feed management and preservation processes is projected to have a positive impact on animal productivity in the future.

## 1. Introduction

A total mixed ration (TMR) can be defined as a complete dietary source that includes forage, cereal grains, protein sources, by-products, minerals, vitamins, and additives, which meets the daily nutritional needs of livestock where they are kept indoors year round [1]. Today, the uniformly distributed balanced nutritional characteristics of TMRs make them popular worldwide due to their beneficial effect on ruminant nutrition, management, and production of ruminants [2].

Farmers use TMRs in their farm for four main reasons: (i) to protect their animals from extreme environmental conditions; (ii) to provide a more constant and consistent feed composition, which also prevents sorting of the feed; (iii) to allow easy records of feed intake and regulation of dry matter intake (DMI); and (iv) to achieve higher milk yield [3]. As indicated previously, TMR also provides a more stable rumen environment with less fluctuation in pH compared to separate feeding of forage and concentrates [4]. On the other hand, one of the biggest obstacles is preventing the sorting of smaller concentrate particles instead of long forage particles in the feed bunker, which increases the risk of subclinical rumen acidosis (SARA) and negatively impacts rumen health [5,6]. A common view among special nutritionists and scientists was that adding water to dry TMRs could reduce this sorting feed behavior due to the adhesive effect of concentrate particles on the larger forage particles. However, the addition of water to TMRs makes them more susceptible to spoilage with increasing environmental temperature [6,7]. Therefore, there seems to be a definite need for a quick and simple method to assess the aerobic stability of the TMR directly from the large feed bunker during feed-out.

Infrared thermography (IRT) is a noninvasive method for measuring radiated heat emitted by an object in real time [8,9]. Compared to conventional methods, this method demonstrates rapidity, dependability, and scalability, with the potential for greater automation through the simultaneous screening of more samples [10]. Moreover, in situations where traditional diagnostic methods have failed, infrared thermography (IRT) is offered as a useful tool for gathering essential information [11]. IRT does, however, have certain limitations and factors that need to be considered when applying it. When employing IRT, it is critical to keep a consistent angle and distance from the subject, as well as to regulate factors such as ambient temperature, wind speed, and direct sunshine [12].

The field of veterinary medicine and animal production has made extensive use of IRT for measuring heat losses in a wide variety of species [13], and it has extended its application area over the past few decades to assess the safety and quality of agricultural products as well as to detect spoilage in silos [14,15,16]. Prior research has also noted the importance of detecting the early stages of decomposition in silos in order to achieve a more efficient preservation process [17,18]. The ability to read the temperature distribution remotely without direct physical contact makes this method more appropriate for use on farms. Furthermore, mapping the surface temperature of the silo or feed bunker allows feeding management to minimize feed loss. Although some research has been carried out to monitor temperature differences during air exposure using the IRT method, no studies have been found that directly focus on the correlation between nutritional composition and the IRT values, and factors affecting these values [1,15,16]. Hence, this study aims at addressing these knowledge gaps in the state of the art by using principal components and multifactor analysis. This is to reveal which factors play a crucial role in the aerobic stability of water-added TMRs. Study objectives included determining what factors could cause a rise in temperature when the IRT method is used in the field.

## 2. Materials and Methods

### 2.1. Experimental Design and Treatments 

The nutritional requirements of lactating dairy cows that produce 35 kg/d of milk specified by the NRC [19] were considered in formulating the TMR. The ingredients and chemical composition of the TMR are shown in Table 1. 

Corn silage was already fermented with silage inoculant Silofit M (CB Ideal Animal Nutrition & Health, Afyonkarahisar, Turkey) with an application rate of 0.05 mL/kg (according to the manufacturer’s recommendation) in cylindrical bales wrapped with seven layers of plastic film at least 45 days before preparation of the TMR. The ingredients were mixed in a stationary wagon (Mascus RMH VS model, 22 m^3^) for 10 min without water addition. The 90 kg of TMR was brought to the lab as soon as possible, where it was split into four equal parts for use in the experiments. Then, 3.75 kg of TMR was spread out thinly over a clean nylon cover and water added to get the moisture content to 40%, 45%, 50%, or 55%. In order to attain the targeted moisture level, the precise quantity of water needed was determined through calculation. Following that, the water was sprayed onto the mixture using a hand spray at the prescribed application rate. The mixture was then homogenized using a gloved hand. Triplicate samples of the TMR were prepared and then exposed to air for 48 h at two different storage temperatures (24 °C and 30 °C) in separated air-conditioned laboratory rooms with a relative humidity of 55%. Subsamples were taken from the midpoint of each TMR at hours 0, 2, 6, 12, 24, and 48 of air exposure for further chemical and microbiological analysis.

### 2.2. Recording of IRT Values of TMR

The central temperature (CT), mean temperature (MT), and maximum temperature difference (MTD) of the TMR were recorded in triplicate with the T200 IR model thermal camera at each time (0, 2, 6, 12, 24, and 48 h) before subsampling and evaluated with ThermaCAM SmartView Classic 4.3 software (Fluke Corporation, Washington, DC, USA). Prior to collecting the image, the camera was subjected to a 10-min acclimatization phase, as recommended by Orman and Endres [20]. We maintained a standard distance of 0.5 m between the thermal camera and the target TMR in the evaluated areas, along with markings to ensure a 0.5-m separation between the camera and the target TMR. The laboratory’s ambient temperature was 22 °C with a relative humidity of 55% when the IRT values of the TMRs were measured. In accordance with the manufacturer’s instructions, the emissivity value was adjusted to 0.98, and the thermograph resolution was calibrated to ambient temperature and humidity.

### 2.3. Chemical and Microbiological Analyses

The pH of each TMR subsample was measured with a pH meter (WTW-inoLab pH 730) after extracting a representative 20 g of fresh TMR sample mixed gently in 180 mL of distilled water at room temperature for 1 h. The dry matter (DM) was determined by drying at 60 ± 2 °C in an air-forced oven for 48 h. The crude protein (CP) and ether extract (EE) contents of the TMR were analyzed according to methods 976.05 and 920.39, respectively, of AOAC [21]. The ash content was determined by incinerating the TMR samples in a muffle furnace at 550 °C for 3 h. Neutral detergent fiber (NDF) and acid detergent fiber (ADF) of the TMR subsamples were determined according to Van Soest et al. [22]. Water-soluble carbohydrates (WSC) of the samples were analyzed according to Esen et al. [23] by using a 0.2% anthrone reagent. The lactic acid (LA) content of the samples was determined using a spectrophotometric method previously described by Koc and Coskuntuna [24]. The LAB (lactic acid bacteria) and yeast of the samples were determined using MRS (De Man, Rogosa and Sharpe) and PDA (potato dextrose agar) agar (Merck, Darmstadt, Germany), according to Bağcık et al. [25] and presented on a wet basis.

### 2.4. Statistical Analysis

Data were adjusted using the PROC MIXED procedure [26] for the fixed effects of moisture level (40, 45, 50, and 55%), storage temperature (24 and 30 °C), time (0, 2, 6, 12, 24, and 48 h), and the interaction between these effects. The correlogram was drawn for visualization in the ‘*corrplot*’ package to assess the strength and direction of association between storage temperature, time, fermentation characteristics, microbial mix, chemical compositions, cell wall components, and TMR IRT values [27]. The ‘*psych*’ package in R was used to perform a loading vector analysis in order to enhance the comprehension of the effect of each variable on aerobic stability. Prior to the loading vector analysis, the highly correlated variable MT-CT was eliminated. From the varimax rotation results, only loading vectors with a sum of squares value greater than one were tabulated, and variables were considered associated with a specific component only if their loading vector exceeded 0.70. Additionally, the “*ggplot2*” package was used to visualize the biplot order, based on principal component analysis (PCA), for TMRs with varying moisture levels. Then, all variables were categorized into 3 quantitative groups (moisture level, storage temperature, time) and 5 qualitative groups (IRT values, chemical composition, cell wall component, microbial composition, fermentation characteristics) to perform multiple factor analysis (MFA) in R using the ‘*factoextra*’ and ‘*FactoMineR*’ packages [28].

## 3. Results

The fermentation characteristics of the TMR during air exposure are summarized in Table 2. Moisture level and time significantly affected pH, DM, WSC, and LA content (*p* < 0.001). Except for WSC (*p* < 0.05), the temperature was another important factor that affected the other parameters at the same significance level (*p* < 0.001). There was an increase in DM content of all TMR groups due to ambient temperature and evaporation during 48 h of air exposure (*p* < 0.001). However, no interaction effect was detected between moisture level, storage temperature, and time (*p* > 0.05). Interestingly, the expected increase in LA content was not observed after the sharp decrease in pH and WSC after 24 h of air exposure.

Table 3 provides the summary statistics for the LAB and the yeast count of the TMR. Unlike storage temperature (*p* > 0.05), the moisture level of the TMR and air exposure time were significant factors for the LAB and yeast count (*p* < 0.001). The interactions between moisture level, storage temperature, and air exposure time were also significant (*p* < 0.001 except storage temperature × air exposure time, *p* < 0.01). A closer inspection of this figure also shows a decreasing trend in the LAB count in all TMR groups except TMR55 after 12 h. On the other hand, the yeast count of the TMR groups fluctuated during the air exposure time. The highest yeast count was observed in the TMR groups after 2 h of air exposure, except TMR40.

NDF was the only parameter not affected by moisture level, storage temperature, air exposure time, and their interactions (*p* > 0.005; Table 4). Moisture levels significantly affect the content of CP, EE, and ADF contents of TMR. Unlike NDF, the ADF content of the TMR was significantly affected by moisture level, air exposure time, and their interactions (*p* < 0.001). The interactions between moisture levels × air exposure time and air exposure time × storage temperature were also significant (*p* < 0.05). Although the ash content of the TMR was affected only by the air exposure time (*p* < 0.01), CP and EE were significantly affected by the moisture level, the storage temperature, and the air exposure time (*p* < 0.001). Furthermore, the CP and EE content of all TMR groups began to decrease after 24 h of exposure to air in parallel with the reduced WSC content of TMR.

The relationship between fermentation characteristics, microbial and chemical compositions, cell wall components, and TMR IRT values was examined using the Pearson correlation coefficient and is presented in Figure 1. From these data, we can see that IRT values (MT, CT, MTD) were significantly correlated with LAB, WSC, and pH (*p* < 0.001). The highest positive correlation (*r* = 0.98) was observed between MT and CT, which means that they can be used interchangeably in field applications. On the contrary, the highest negative correlation was observed between WSC and time (*r* = −0.88). On the other hand, no correlation was detected between ST and the other parameters, except MT and CT (*p* > 0.05).

Five loading vectors emerged after the Varimax rotation, which explains 67.4% of the total variability of original variables (Table 5). It is apparent from this table that the first two components of PCA explain 44.2% of the total variation. 

Further statistical tests revealed that the variable loadings on PC1 were related to the IRT values (CT and MTD) and the fermentation characteristics (pH, LAB, and WSC), and time explained 32.0% of the total variation. PC2 was characterized by the negative relationship between the DM and LA content of the TMR, accounting for 12.3% of the overall variation. Although EE and ST were the main components of PC3 and PC5, NDF could be considered a main component of PC4 due to its closer value to 0.70. The biplot ordering using PCA of water-added TMR groups is also presented in Figure 2. 

Although there was no clear distinction between the trial groups, only TMR40 and TMR55 differed from each other, except for the content of EE and LAB. Furthermore, the most striking result that emerged from the MFA results was that fermentation characteristics, which are directly affected by chemical composition and moisture level, are the key point in determining the quality of the TMR (Figure 3). Moreover, it is possible to obtain some essential foresight, such as related parameters for the temperature increase, by using IRT technology under field conditions due to the high correlation between IRT values and fermentation characteristics.

## 4. Discussion

The microbiological and chemical composition of feed over the entire feeding period requires the expertise of skilled specialists, expensive equipment, and a specialized laboratory. Our field-based IRT pilot study results show that this technique has the potential to improve silage management procedures by determining the level of degradation occurring during the onset of aerobic stability [16]. Furthermore, previous research has shown that any significant changes in surface temperature during the decomposition of feed can be quickly detected by monitoring surface temperatures at various locations [17,18].

In the realm of water addition to dry TMR, numerous reports have been published, focusing on its effects on feed intake, feed sorting behavior, and the health and performance of dairy cows [5,6,7,29,30]. Data from these studies suggest that it is possible to reduce sorting behavior against the longest particles and increase the consumption of dry matter by adding water to a dry diet. Another important finding is that this cost-effective management practice can help to promote a healthy rumen in early lactation [30]. There is, however, a lack of information in the existing literature concerning the effects of water addition on feed temperature and nutritional properties in the bunk over the course of the day. In one of the rare studies, Felton and DeVries [7] scanned the feed temperature using IRT technique and underlined its feasibility due to the strong correlation between the feed temperature recorded in the thermal image camera and the temperature probe. Nonetheless, this study did not provide any insights into the relationship between IRT values and the chemical and microbial composition. Therefore, the present study provides the first comprehensive assessment of the relationship between IRT values and feed characteristics of TMRs with different moisture levels on a laboratory scale.

In previous research, it was demonstrated that certain atmospheric conditions, such as ambient temperature, relative humidity, and wind speed, may interfere with the acquisition of thermal camera signals. Therefore, in the current study, measures were taken to reduce the impact of these factors on data by manipulating the temperature, relative humidity, and airflow in the laboratory setting. However, the main weakness of this study was the paucity of air exposure time due to the stimulation of the time interval between the preparation and consumption of the TMR in farm conditions. 

As has already been noted, the main purpose of using LAB strains alone or in combination with fibrolytic enzymes, sugar sources, and/or organic acid is to guarantee silage fermentation [31,32]. A closer inspection of Table 2 shows that all TMRs had higher WSC (>50 g/kg DM) content at initial air exposure, which can allocate more substrate for LA production [33]. However, the initial LAB population of all TMRs was far from suppressing yeast count [34,35]. Furthermore, an increase in dry matter loss, especially after 24 h of air exposure, could be related to yeast activity that uses fermentation products and LA as primary energy sources. The reduced WSC content of all TMR groups strengthens this assumption. Another inhibitory effect on the LAB population might be the added sodium bicarbonate salts in the TMRs related to ruminal acidosis. However, the considerably low LA content and the insufficient LAB population cannot explain the decreasing trend in pH in this study. According to Holzer et al. [36] and Auerbach and Nadeau [37], it can be attributed to the novel metabolic pathway of heterofermentative *Lentilactobacillus buchneri* (formerly known as *Lactobacillus buchneri*) from LA to acetic acid (AA) and 1,2-propanediol or secondary oxidation of LA to AA during air exposure.

It is well known from previous studies that dairy cows require forage fiber in their diet not only to maintain rumen function but also to maximize milk production. Therefore, an accurate evaluation of the physical characteristics of the TMR and its cell wall fraction, mainly NDF and its degradability, is much more important for the formulation of the diet and lactation performance of dairy cows [38,39]. Contrary to previously published studies by Felton and DeVries [7], the addition of water did not affect the NDF values of the TMRs in the current study. The relatively low LAB population and activity resulted in a higher ADF content on the increase in the DM content of the TMRs. These results are also similar to those reported by Wang et al. [40].

A recently performed meta-analysis, which examined the effect of the inoculant rate of obligate homofermentative and facultative heterofermentative LAB and obligate heterofermentative LAB on silage quality, presents considerable correlation among response variables of both forage- and legume-based silage, one of the most important ingredients in TMRs, and provides much deeper knowledge [41]. In that study, a positive correlation was reported between pH and ADF and a negative correlation between pH and LA content in forage-based silage. This differs from the findings presented here. Furthermore, small but significant negative correlations between ash and fermentation characteristics (except LA) were observed in the TMR groups in the present study. However, no clear increasing tendency for ash content through the TMR groups was seen in terms of moisture level. It might be related to the aerobic deterioration of TMRs, which results in loss of organic matter. These results are similar to those reported by de Oliveira et al. [42]. 

Gallo et al. [43] argued that it is possible to identify the main sources of the variation in the original variables and to obtain more information than the quantitative measurements themselves by using a multivariate statistical technique. The authors also proposed that the principal components should be related to the chemical composition, fermentation characteristics, and digestibility for silage studies. Furthermore, there is no doubt that additional categories should be added to the factor analysis, considering the objective of the study. Therefore, all variables studied were categorized into three quantitative (moisture level, storage temperature, time) and five qualitative groups (IRT values, chemical composition, cell wall component, microbial composition, fermentation characteristics) to perform MFA analysis. In the previous section, it was stated that the fermentation characteristics located in Dims 1 and 2 had higher contributions to the total variance and played a significant role in preventing aerobic deterioration of the TMR. The moisture level and chemical composition of the TMR were the other categories that contributed the most. These results are similar to those obtained by Koç et al. [44], who reported that the moisture level, the type of ensiled plant, and its parts are more important than biological treatments for adequate forage conservation.

## 5. Conclusions

In this research, the IRT technique is demonstrated to be an effective method of evaluating the aerobic stability of water-added TMRs during feed-out. According to the findings, IRT values, such as CT and MTD, are substantially influenced by fermentation parameters, including pH, WSC content, and LAB count. These results highlight the possibility of preventing economic losses and optimizing feeding methods by identifying portions with higher CT or MTD data using IRT measurements just before feeding dairy cows in field conditions. Moreover, considering the substantial influence of fermentation characteristics on overall variances, implementing such manipulations could be essential in enhancing the aerobic stability of high-moisture TMRs. Further research and implementation of IRT in feed management and preservation processes could have a positive impact on animal productivity.

## Figures and Tables

**Figure 1 animals-13-02225-f001:**
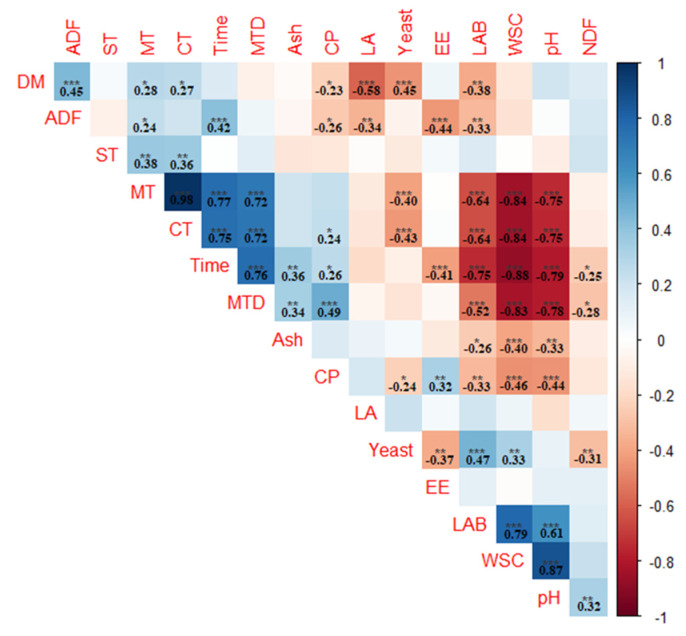
The Pearson’s correlation coefficients of fermentation characteristics, microbial and chemical compositions, cell wall components, and infrared thermography values of the TMR with different moisture levels. Blue and red squares indicate positive and negative correlations, respectively, and the empty cases refer to non-significant correlations. ADF: acid detergent fiber; CP: crude protein; CT: central temperature; DM: dry matter; EE: ether extract; LA: lactic acid; LAB: lactic acid bacteria; MT: mean temperature; MTD: maximum temperature differences; NDF: neutral detergent fiber; ST: storage temperature; WSC: water-soluble carbohydrates; *: *p* < 0.05, **: *p* < 0.01, ***: *p* < 0.001.

**Figure 2 animals-13-02225-f002:**
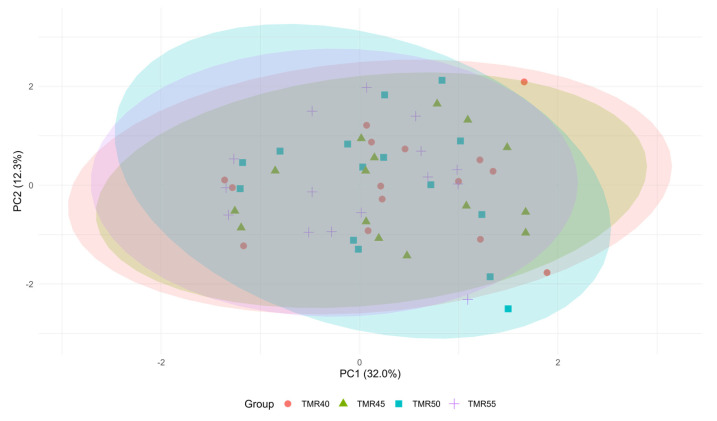
Biplot ordering using principal component analysis of TMR with different moisture levels. ADF: acid detergent fiber; CP: crude protein; CT: central temperature; DM: dry matter; EE: ether extract; LA: lactic acid; LAB: lactic acid bacteria; MT: mean temperature; MTD: maximum temperature differences; NDF: neutral detergent fiber; ST: storage temperature; WSC: water-soluble carbohydrates.

**Figure 3 animals-13-02225-f003:**
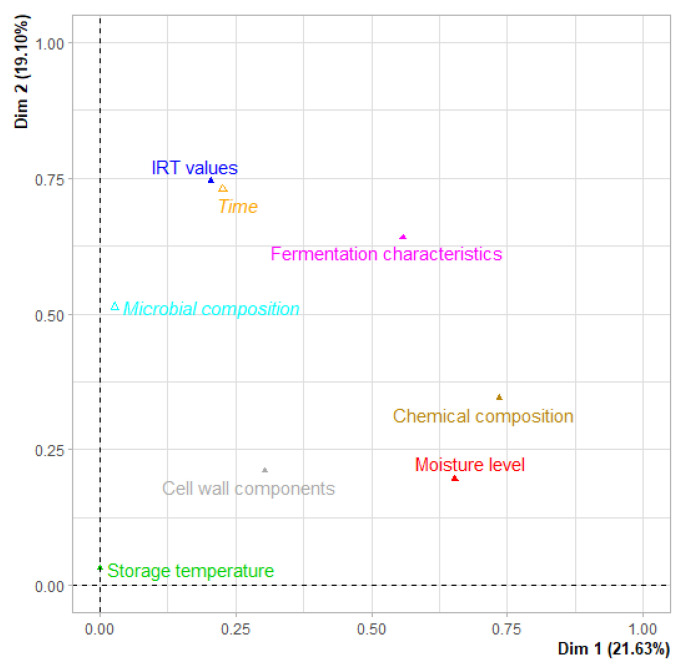
Multiple factor analysis results of TMR with different moisture levels.

**Table 1 animals-13-02225-t001:** Ingredients and chemical composition of the TMR.

Item	TMR
Ingredients (g/kg of DM)	
Corn silage ^a^	241.1
Corn grain (high moisture)	180.6
DDGS (corn)	53.6
Barley	33.7
Sunflower meal	15.0
Alfalfa hay	140.3
Canola meal	50.3
Sugar beet pulp	5.7
Wheat straw	22.7
Sunflower grain	9.2
Soybean peel	9.3
Cottonseed	58.9
Rice bran	36.8
Wheat bran (fine)	24.3
Molasses	8.9
Orange pulp	34.9
Maceration water	51.2
Marble dust	6.2
Vit + Min premix ^b^	4.2
Ecomass	4.2
Buffer (Sodium bicarbonate)	3.0
Salt	2.1
Potassium carbonate	2.0
OmniGen AF	1.4
Toxin binder	0.4
Chemical composition (g/kg of DM)	
DM	609.8 ± 8.05
CP	144.6 ± 1.34
EE	40.4 ± 0.18
Ash	82.8 ± 0.75
NDF	378.2 ± 0.10
ADF	261.8 ± 6.71

DM: dry matter; CP: crude protein; EE: ether extract; NDF: neutral detergent fiber; ADF: acid detergent fiber. ^a^ Supplied by CB Ideal Animal Nutrition & Health Ltd. (Afyonkarahisar, Turkey), including the ingredients *Lactobacillus casei* SSNGY 4440 (2 × 10^8^ cfu/mL), *Lactobacillus plantarum* ATCC 8014 (1 × 10^8^ cfu/mL), *Lactobacillus brevis* IFA 92 (1 × 10^8^ cfu/mL), *Pediococcus acidilactici* 33-11 NCIMB 30085 (1 × 10^8^ cfu/mL), *Lentilactobacillus buchneri* CCM 1819 (1 × 10^8^ cfu/mL), and *Pediococcus pentosaceus* NCIMB 30168 (1 × 10^8^ cfu/mL). ^b^ Supplied by Rasyonel Tarımsal Besicilik Ltd. (Inece, Kırklareli, Turkey), including the ingredients Vitamin D3 (2749 IU/kg), Vitamin A (13743 IU/kg), FeSO_4_.7H_2_O (46 mg/kg), CuSO_4_.5H_2_O (9 mg/kg), Ca(IO_3_)_2_, ZnO (46 mg/kg), Na_2_SeO_3_ (0.137 mg/kg), and CoSO_4_.5H_2_O (0.137 mg/kg).

**Table 2 animals-13-02225-t002:** Changes in the pH, dry matter (DM), water-soluble carbohydrates (WSC), and lactic acid (LA) content of the TMR with different moisture levels.

Item	Time	24 °C				30 °C							*p*-Values				
		40%	45%	50%	55%	40%	45%	50%	55%	SEM	M	ST	T	MxST	MxT	STxT	MxSTxT
pH	0	4.91 ^abc^	4.92 ^ab^	4.85 ^a–e^	4.82 ^b–i^	4.91 ^abc^	4.92 ^ab^	4.85 ^a–e^	4.82 ^b–i^	0.02	<0.0001	0.0007	<0.0001	0.0605	0.0581	0.0138	0.3445
	2	4.94 ^a^	4.89 ^a–d^	4.86 ^a–e^	4.85 ^a–e^	4.83 ^b–i^	4.85 ^a–e^	4.81 ^d–l^	4.83 ^b–g^								
	6	4.86 ^a–e^	4.83 ^b–h^	4.82 ^c–j^	4.82 ^c–j^	4.84 ^b–f^	4.84 ^b–f^	4.82 ^c–j^	4.82 ^c–j^								
	12	4.79 ^d–n^	4.80 ^d–n^	4.77 ^e–o^	4.80 ^d–m^	4.79 ^d–n^	4.81 ^d–k^	4.78 ^e–n^	4.77 ^e–o^								
	24	4.83 ^b–i^	4.78 ^e–n^	4.71 ^mno^	4.72 ^k–o^	4.73 ^i–o^	4.77 ^e–o^	4.72 ^j–o^	4.70 ^mno^								
	48	4.74 ^f–o^	4.74 ^g–o^	4.71 ^mno^	4.71 ^l–o^	4.73 ^h–o^	4.72 ^k–o^	4.70 ^no^	4.68 ^o^								
DM	0	609.8 ^b^	562.7 ^c^	511.5 ^def^	462.9 ^h^	609.8 ^b^	562.7 ^c^	511.5 ^def^	462.9 ^h^	4.64	<0.0001	<0.0001	<0.0001	0.3098	0.9971	0.6230	0.9986
	2	618.8 ^ab^	566.1 ^c^	513.0 ^de^	466.1 ^gh^	622.9 ^ab^	568.8 ^c^	520.2 ^d^	478.6 ^gh^								
	6	620.7 ^ab^	566.4 ^c^	513.4 ^de^	469.0 ^gh^	623.7 ^ab^	570.7 ^c^	522.3 ^d^	480.7 ^gh^								
	12	626.2 ^ab^	567.0 ^c^	519.1 ^d^	470.3 ^gh^	626.9 ^ab^	574.3 ^c^	529.0 ^d^	482.2 ^gh^								
	24	628.9 ^ab^	578.3 ^c^	521.9 ^d^	475.6 ^gh^	636.1 ^ab^	576.6 ^c^	530.9 ^d^	484.8 ^fgh^								
	48	630.1 ^ab^	579.7 ^c^	528.3 ^d^	481.0 ^gh^	639.6 ^a^	581.0 ^c^	532.2 ^d^	490.3 ^efg^								
WSC	0	80.2 ^d–g^	89.6 ^a^	79.6 ^d–h^	73.2 ^hij^	80.2 ^d–g^	89.6 ^a^	79.6 ^d–h^	73.2 ^hij^	1.17	<0.0001	0.0200	<0.0001	<0.0001	<0.0001	<0.0001	<0.0001
	2	85.1 ^a–d^	80.4 ^d–g^	85.1 ^a–d^	82.2 ^b–e^	88.0 ^ab^	70.9 ^j^	77.6 ^e–j^	74.4 ^f–j^								
	6	81.0 ^c–f^	87.7 ^abc^	71.0 ^j^	88.0 ^ab^	72.5 ^ij^	71.8 ^ij^	76.4 ^e–j^	87.9 ^ab^								
	12	89.3 ^a^	55.6 ^k^	75.5 ^e–j^	78.6 ^d–i^	91.7 ^a^	87.5 ^abc^	73.8 ^g–j^	71.6 ^j^								
	24	18.8 ^lmn^	14.8 ^l–o^	10.3 ^o^	20.4 ^l^	14.4 ^l–o^	16.9 ^l–o^	12.5 ^mno^	19.2 ^lm^								
	48	14.3 ^l–o^	17.6 ^l–n^	3.4 ^p^	15.2 ^l–o^	12.0 ^no^	10.8 ^o^	12.6 ^mno^	12.8 ^mno^								
LA	0	6.1 ^e–j^	6.4 ^e–j^	7.7 ^d–i^	10.9 ^abc^	6.1 ^e–j^	6.4 ^e–j^	7.7 ^d–i^	10.9 ^abc^	0.48	<0.0001	0.4506	<0.0001	<0.0001	<0.0001	<0.0001	<0.0001
	2	7.3 ^d–j^	8.7 ^b–e^	8.2 ^c–f^	7.0 ^d–j^	7.3 ^d–j^	8.2 ^c–f^	6.3 ^e–j^	6.8 ^e–j^								
	6	7.5 ^d–i^	7.1 ^d–j^	11.5 ^a^	8.1 ^def^	6.7 ^e–j^	7.2 ^d–j^	6.0 ^e–j^	11.7 ^a^								
	12	6.3 ^e–j^	11.8 ^a^	5.3 ^g–j^	7.2 ^d–j^	8.2 ^c–f^	6.0 ^e–j^	11.4 ^ab^	8.2 ^c–f^								
	24	5.3 ^hij^	8.1 ^d–h^	9.7 ^a–d^	11.1 ^ab^	7.1 ^d–j^	4.6 ^j^	7.9 ^d–i^	8.1 ^d–h^								
	48	5.2 ^ij^	8.0 ^d–i^	5.6 ^f–j^	6.0 ^e–j^	8.1 ^d–g^	6.8 ^e–j^	7.5 ^d–i^	8.2 ^c–f^								

M: moisture level; ST: storage temperature; T: air exposure time; M × ST: interaction between moisture level and storage temperature; M × T: interaction between moisture level and air exposure time; M × ST × T: interaction between moisture level, storage temperature, and air exposure time. In each item, values with different letters (a–p) are statistically different.

**Table 3 animals-13-02225-t003:** Changes in lactic acid bacteria (LAB) and yeast count of the TMR with different moisture levels.

Item	Time	24 °C				30 °C							*p*-Values				
		40%	45%	50%	55%	40%	45%	50%	55%	SEM	M	ST	T	MxST	MxT	STxT	MxSTxT
LAB	0	2.47 ^ab^	2.25 ^a–e^	2.30 ^a–d^	2.42 ^abc^	2.47 ^ab^	2.25 ^a–e^	2.30 ^a–d^	2.42 ^abc^	0.10	<0.0001	0.5523	<0.0001	0.0002	<0.0001	<0.0001	<0.0001
	2	2.13 ^a–g^	1.91 ^b–j^	1.88 ^c–k^	1.73 ^e–k^	1.76 ^d–k^	1.68 ^f–l^	2.05 ^a–h^	1.58 ^g–m^								
	6	1.53 ^h–m^	1.90 ^c–j^	2.17 ^a–f^	2.03 ^a–h^	2.02 ^a–i^	1.74 ^e–k^	1.98 ^a–i^	2.11 ^a–g^								
	12	1.84 ^d–k^	1.93 ^b–j^	1.70 ^e–k^	2.54 ^a^	1.77 ^d–k^	1.82 ^d–k^	1.98 ^a–i^	1.52 ^h–m^								
	24	1.83 ^d–k^	1.33 ^klm^	1.49 ^h–m^	1.97 ^a–i^	1.39 ^j–m^	1.70 ^e–k^	1.76 ^d–k^	1.92 ^b–j^								
	48	1.11 ^m^	1.13 ^lm^	1.03 ^m^	1.97 ^b–i^	1.46 ^i–m^	1.50 ^h–m^	1.72 ^e–k^	2.12 ^a–g^								
Yeast	0	3.18 ^a–g^	3.06 ^c–m^	3.09 ^b–k^	3.14 ^a–i^	3.18 ^a–g^	3.06 ^c–m^	3.09 ^b–k^	3.14 ^a–i^	0.04	<0.0001	0.7214	<0.0001	0.2337	<0.0001	0.0018	0.0003
	2	2.85 ^no^	3.08 ^b–l^	3.18 ^a–g^	3.23 ^a–d^	2.81 ^o^	3.15 ^a–i^	3.34 ^a^	3.26 ^abc^								
	6	3.03 ^d–n^	3.10 ^b–j^	3.22 ^a–d^	3.16 ^a–h^	3.08 ^b–l^	3.04 ^d–n^	3.09 ^b–l^	3.14 ^b–i^								
	12	2.90 ^k–o^	2.99 ^f–o^	3.09 ^b–l^	3.05 ^d–n^	2.89 ^l–o^	3.00 ^e–o^	3.00 ^e–o^	3.28 ^ab^								
	24	2.95 ^i–o^	2.99 ^g–o^	2.98 ^h–o^	3.19 ^a–f^	2.90 ^k–o^	2.87 ^mno^	2.92 ^j–o^	3.09 ^b–k^								
	48	3.12 ^b–j^	3.01 ^e–o^	3.05 ^d–n^	3.20 ^a–e^	2.98 ^g–o^	3.09 ^b–k^	3.12 ^b–j^	3.22 ^a–d^								

M: moisture level; ST: storage temperature; T: air exposure time; M × ST: interaction between moisture level and storage temperature; M × T: interaction between moisture level and air exposure time; M × ST × T: interaction between moisture level, storage temperature, and air exposure time. In each item, values with different letters (a–o) are statistically different.

**Table 4 animals-13-02225-t004:** Changes in crude protein (CP), ether extract (EE), ash, neutral detergent fiber (NDF), and acid detergent fiber (ADF) content of the TMR with different moisture levels.

Item	Time	24 °C				30 °C							*p*-Values				
		40%	45%	50%	55%	40%	45%	50%	55%	SEM	M	ST	T	MxST	MxT	STxT	MxSTxT
CP	0	144.6 ^cde^	145.9 ^cd^	143.6 ^de^	144.5 ^cde^	144.6 ^cde^	145.9 ^cd^	143.6 ^de^	144.5 ^cde^	0.58	<0.0001	0.0005	<0.0001	<0.0001	<0.0001	<0.0001	<0.0001
	24	144.8 ^cde^	150.8 ^a^	151.5 ^a^	150.8 ^a^	150.5 ^ab^	149.5 ^ab^	147.4 ^bc^	151.3 ^a^								
	48	140.0 ^f^	151.7 ^a^	152.1 ^a^	149.8 ^ab^	145.7 ^cd^	142.2 ^ef^	143.9 ^de^	150.3 ^ab^								
EE	0	40.4 ^f^	36.4 ^g^	35.4 ^h^	35.2 ^h^	40.4 ^f^	36.4 ^g^	35.4 ^h^	35.2 ^h^	0.18	<0.0001	<0.0001	<0.0001	<0.0001	<0.0001	<0.0001	<0.0001
	24	43.5 ^cd^	46.1 ^b^	41.8 ^e^	42.5 ^de^	44.4 ^c^	43.4 ^d^	43.1 ^d^	48.1 ^a^								
	48	31.0 ^ijk^	30.3 ^kl^	31.3 ^ij^	31.8 ^i^	30.4 ^jkl^	29.8 ^l^	34.8 ^h^	30.3 ^jkl^								
Ash	0	82.8	81.5	82.9	83.9	82.8	81.5	82.9	83.9	1.07	0.2877	0.1944	0.0035	0.4221	0.1898	0.6492	0.1402
	24	84.1	84.6	84.2	85.6	85.1	84.4	83.3	82.4								
	48	85.4	84.4	84.9	85.4	82.4	86.5	81.7	85.9								
NDF	0	378.2	383.0	382.9	387.9	378.2	383.0	382.9	387.9	8.49	0.3857	0.0590	0.0743	0.2717	0.2024	0.3812	0.0895
	24	387.3	383.5	374.3	351.2	387.2	386.6	381.7	375.8								
	48	356.5	386.1	356.5	370.4	394.2	368.0	382.7	370.1								
ADF	0	261.8 ^b–e^	298.7 ^ab^	273.9 ^b–e^	244.7 ^e^	261.8 ^b–e^	298.7 ^ab^	273.9 ^b–e^	244.7 ^e^	7.25	<0.0001	0.2411	<0.0001	0.0129	<0.0001	0.0209	0.2295
	24	287.7 ^a–d^	287.7 ^a–d^	263.6 ^b–e^	254.9 ^cde^	267.3 ^b–e^	253.6 ^de^	262.6 ^b–e^	249.2 ^de^								
	48	318.4 ^a^	293.3 ^abc^	267.0 ^b–e^	288.2 ^a–d^	324.4 ^a^	273.7 ^b–e^	300.8 ^ab^	287.2 ^a–d^								

M: moisture level; ST: storage temperature; T: time; M × ST: interaction between moisture level and storage temperature; M × T: interaction between moisture level and air exposure time; M × ST × T: interaction between moisture level, storage temperature, and air exposure time. In each item, values with different letters (a–l) are statistically different.

**Table 5 animals-13-02225-t005:** Loading vectors of original variables of the TMR with different moisture levels ^1^.

Original Variable	PC1	PC2	PC3	PC4	PC5
ST	0.026	−0.003	0.006	0.123	**0.877**
Time	**0.922**	0.195	0.322	−0.133	−0.020
CT	0.053	0.057	−0.275	−0.087	−0.025
MT	**0.828**	0.234	−0.015	0.147	0.395
MTD	**0.837**	−0.021	−0.122	−0.168	0.127
DM	0.021	**0.836**	0.028	0.249	0.006
pH	**−0.913**	0.209	0.050	0.141	−0.097
Yeast	−0.244	−0.391	0.426	−0.590	−0.040
LAB	**−0.763**	−0.351	0.010	−0.180	0.235
LA	0.009	**−0.735**	−0.001	0.123	−0.029
WSC	**−0.976**	−0.115	0.131	−0.021	0.020
CP	0.460	−0.251	−0.444	0.063	−0.163
Ash	0.371	−0.092	0.038	−0.075	−0.137
EE	−0.125	−0.024	**−0.734**	0.239	0.056
ADF	0.170	0.471	0.601	0.191	−0.087
NDF	−0.233	0.001	0.118	0.658	0.132
SS loadings	5.120	1.961	1.511	1.103	1.084
Proportion (%)	32.0	12.3	9.4	6.9	6.8
Cumulative (%)	32.0	44.3	53.7	60.6	67.4

^1^ Variables with loading vectors higher than 0.70 were considered loading on a specific component and highlighted in bold. ST: storage temperature; CT: central temperature; MT: mean temperature; MTD: maximum temperature differences; DM: dry matter; LAB: lactic acid bacteria; LA: lactic acid; WSC: water-soluble carbohydrates; CP: crude protein; EE: ether extract; ADF: acid detergent fiber; NDF: neutral detergent fiber; SS: sum of square.

## Data Availability

Not applicable.

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
