# Peer review of "Infrared Thermography Assessment of Aerobic Stability of a Total Mixed Ration: An Innovative Approach to Evaluating Dairy Cow Feed"

_animals, 2023, doi:10.3390/ani13132225_

Round 1
Reviewer 1 Report
This study provided comprehensive analysis about the effect of moisture content on the aerobic stability of total mixed ration. The methodology is described clearly and results are presented completely.
Few minor issue can be addressed:
Table 1. for the chemical composition, can you provide the standard deviation of the result to improve the data comparability with other study. In addition, can you describe the method of chemical composition analysis for reproducibility?
Table 2 indicates that the MT and CT are taking large portion in PC1, however Figure 4 also shows MT and CT are highly correlated to each other, wondering can you do a loading vectors analysis after removing the correlation among the variables for better understanding the effect of each variables on aerobic stability?
Author Response
Response to Reviewer 1 Comments
This study provided comprehensive analysis about the effect of moisture content on the aerobic stability of total mixed ration. The methodology is described clearly and results are presented completely.
Please accept our sincere thanks for your insightful comments and suggestions.
Point 1: Table 1. for the chemical composition, can you provide the standard deviation of the result to improve the data comparability with other study. In addition, can you describe the method of chemical composition analysis for reproducibility?
Response 1: In accordance with your recommendations, standard deviation has been added to the results of TMR's chemical analysis. The methods used for chemical analyses in our study were described in our original article, which was submitted. However, if you are interested, you can refer to our revised article L141-155 for more details.
Point 2: Table 2 indicates that the MT and CT are taking large portion in PC1, however Figure 4 also shows MT and CT are highly correlated to each other, wondering can you do a loading vectors analysis after removing the correlation among the variables for better understanding the effect of each variables on aerobic stability?
Response 2: As per your suggestion, the Table in the original text was reorganized by excluding the correlation between MT and CT. The modifications made have been presented in Table 5 and Figure 2 in the revised article. Additionally, you can refer to the updated method description in L162-169 for further details on the changes made.

Reviewer 2 Report
General consideration: The general purpose of the article is defined both in the abstract and in the simple summary.
The aim of this study was to evaluate the quality of high-moisture total mixed rations (TMR) for livestock feed and explore possible manipulations that can improve their fermentation characteristics and stability using infrared thermography.
The keywords that are not fully representative of the study must be reviewed (reading them would seem more like a work of mathematics). Information is missing in the materials and methods section. Figures that create too much confusion need to be reviewed.
The results are too focused on the mathematical aspect and little on the effects on animal nutrition.
1. Introduction
Extend the part about the IRT method (line 68-80) since, according to the title, it is the most important part.
2. Materials and Methods
2.1. Experimental Design and Treatments
In table 1 also insert the data relating to the contents of the EE, ash, and ADL.
Line 104: 90 kg?
Line 107: How were the different moisture levels achieved? Describe method.
Line 109: Storage room conditions. Also report relative humidity data
Data relating to the chemical composition of samples with different moisture contents, exposed to different temperatures at different times, are missing. Insert table.
2.2. Recording of IRT values of TMR
In consideration of the title and the objective of the study, the paragraph should be expanded by reporting also the environmental data (for example the relative humidity and the temperature of the room) where the measurements took place. The reading mode must be reported, such as for example the distance at which the video camera was placed and the zero point in the room itself.
3. Results
Report the data of the figures (1-2-3) as a table, otherwise it is too confusing, and leave the figures only to comment on the trends.
Line 246: A figure 6 is cited which does not exist in the paper.
4. Discussion
The paragraph places little emphasis on the use of IRT, please revise the paragraph accordingly.
5. Conclusions
The conclusions need to be reviewed as they relate more to the influence of water addition of the nutritional characteristics of TMR over the course of the day, with a potential use of IRT to measure temperature changes, than to how field use of it may correspond to the correct method.
Author Response
Response to Reviewer 2 Comments
General consideration: The general purpose of the article is defined both in the abstract and in the simple summary.
The aim of this study was to evaluate the quality of high-moisture total mixed rations (TMR) for livestock feed and explore possible manipulations that can improve their fermentation characteristics and stability using infrared thermography.
The keywords that are not fully representative of the study must be reviewed (reading them would seem more like a work of mathematics). Information is missing in the materials and methods section. Figures that create too much confusion need to be reviewed.
The results are too focused on the mathematical aspect and little on the effects on animal nutrition.
The writers greatly appreciate your insightful comments and recommendations. We have strictly followed your suggestions above and below, and have made the necessary changes.
Point 1: Introduction: Extend the part about the IRT method (line 68-80) since, according to the title, it is the most important part.
Response 1: We appreciate you bringing our attention to the right location. It's important to provide a comprehensive overview of the IRT technique to give readers a clear understanding of its relevance to the study. In accordance with your suggestions, the introduction has been expanded to include information about the IRT method. Please refer to L69-81 in the revised manuscript for further information.
Point 2: Experimental Design and Treatments: In table 1 also insert the data relating to the contents of the EE, ash, and ADL.
Response 2: The EE and ash data have been added to Table 1. However, since the ADL analysis was not carried out in this study, we were unable to include the ADL data of TMR in Table 1.
Point 3: Line 104: 90 kg?
Response 3: The statement has been reviewed and ensured that it is not misspelled. Two storage temperatures x four moisture levels x three replicates x 3.75 kilograms equals 90 kilograms.
Point 4: Line 107: How were the different moisture levels achieved? Describe method.
Response 4: We have provided a clearer description of the process of achieving the targeted mositure levels. Please refer to L116-124 in the revised manuscript for further information.
Point 5: Line 109: Storage room conditions. Also report relative humidity data.
Response 5: Additional information was provided regarding the storage room conditions and the recorded relative humidity data.
Point 6: Data relating to the chemical composition of samples with different moisture contents, exposed to different temperatures at different times, are missing. Insert table. Please refer to Table2 -4 in the revised manuscript for further information.
Response 6: In line with your suggestions, the data you requested in the initial version of the article are now presented at the 0th hour within the relevant tables.
Point 7: In consideration of the title and the objective of the study, the paragraph should be expanded by reporting also the environmental data (for example the relative humidity and the temperature of the room) where the measurements took place. The reading mode must be reported, such as for example the distance at which the video camera was placed and the zero point in the room itself.
Response 7: Incorporating your suggestions, the section you mentioned has been expanded and presented in greater detail in the revised version of the article. Please refer to L131-139 in the revised manuscript for further information.
Point 8: Report the data of the figures (1-2-3) as a table, otherwise it is too confusing, and leave the figures only to comment on the trends.
Response 8: According to the suggestions, the requested figures have been arranged and presented in a table form. Please refer to Table 2-4 in the revised manuscript for further information.
Point 9: Line 246: A figure 6 is cited which does not exist in the paper.
Response 9: This has been fixed in the updated article.
Point 10: The paragraph places little emphasis on the use of IRT, please revise the paragraph accordingly.
Response 10: As per your request, the revised manuscript now provides additional arguments and elaborations on IRT. Please refer to L270-297 in the revised manuscript for further information.
Point 11: The conclusions need to be reviewed as they relate more to the influence of water addition of the nutritional characteristics of TMR over the course of the day, with a potential use of IRT to measure temperature changes, than to how field use of it may correspond to the correct method.
Response 11: In accordance with the suggestions provided, the conclusion has been rewritten. Please refer to L353-362 in the revised manuscript for further information.

Round 2
Reviewer 2 Report
The paper has significantly improved, a major shortage remains. Lignin is a structural component present in the cell walls of plants, and its presence in a unifeed can influence the overheating of the food when exposed to high temperatures. When lignin is subjected to high heat, it can undergo chemical reactions that produce volatile compounds, such as guaiacyl acid and syringic acid. Another important aspect to consider concerns the formation of toxic compounds during overheating of food containing lignin. Certain chemical reactions involving lignin can lead to the production of compounds such as polycyclic aromatic hydrocarbons (PAHs). Therefore, a greater presence of lignin in the unifeed can negatively affect the quality of the unifeed administered to the animals, so that not having dosed it also makes the results of the work less efficient because they could have been influenced by it. This is why it was asked to be entered as data.
Author Response
Dear Editor and Esteemed Referee,
We are concerned about the storage temperature parameters specified in our study, notably 24 and 30 °C. It would appear that these temperatures may have been confused with the high temperatures that are generally utilized during the process of turning lignin into biofuels or biochemicals. As is well known, biomass materials can be converted into biofuels through pyrolysis, a thermochemical reaction conducted without oxygen at a specific temperature. In light of the reviewer's previous comments, the production of polycyclic aromatic hydrocarbons (PAHs) poses a significant challenge to the advancement of heat treatment techniques for lignocellulosic biomass [1]. Knowing that the pyrolysis of lignin generates a greater variety and quantity of PAHs than cellulose or hemicellulose, the esteemed referee emphasized the importance of focusing the analysis specifically on lignin.
On the other hand, It's believed that several phases are involved in the production of PAHs from lignin [2-4]. Lignin is broken down first into its principal byproducts like tar and char, and then into PAHs via secondary reactions [2]. Thermal degradation of lignin, as shown by the work of Asmadi et al. [5] begins with the breakage of the relatively weak α-ether and β-ether bonds within the structure of lignin. This cleavage releases guaiacyl and syringyl aromatic compounds. Following that, these aromatics undergo further reactions, resulting in the formation of catechols and pyrogallols, which can then breakdown to produce PAHs [6]. Dorrestijn et al. [7] also found that the connections between lignin units are cleaved at temperatures between 200 and 400 °C, with the α-O-4 ether bond being the weakest.
Nonetheless, we do not believe any of the preceding material is relevant or pertinent to our research. For our study, we selected temperatures of 24 and 30 °C based on their alignment with temperate and hot climates, as these temperatures are commonly experienced by most TMRs in field conditions. It should be noted that 24 and 30 °C do not classify as high temperatures, and we believe this choice appropriately reflects the temperature range encountered in our target environment. Furthermore, unless exceptional conditions occur, the average daily temperature in the research region remains below 30 °C.
The determination of crude fiber in forages, which is an indicator of the total amount of fiber present, is influenced by the combination of ADF and NDF. The ratios of ADF and NDF are also crucial to comprehend since they offer a trustworthy estimation of forage consumption relative to mass [9]. In simpler terms, these metrics aid in predicting how much an animal will eat until satiety—the moment at which its stomach is full and it stops eating—happens. As a result, these parameters were analyzed during the designated intervals.
We hope that after our explanations, your thoughts on our manuscript will be positive.
Cited References:
[1] Zhou, H., Wu, C., Onwudili, J. A., Meng, A., Zhang, Y., & Williams, P. T. (2014). Polycyclic aromatic hydrocarbon formation from the pyrolysis/gasification of lignin at different reaction conditions. Energy & fuels, 28(10), 6371-6379.
[2] Sharma, R. K., & Hajaligol, M. R. (2003). Effect of pyrolysis conditions on the formation of polycyclic aromatic hydrocarbons (PAHs) from polyphenolic compounds. Journal of Analytical and Applied Pyrolysis, 66(1-2), 123-144.
[3] Tamburini, D., Ribechini, E., & Colombini, M. P. (2016). New markers of natural and anthropogenic chemical alteration of archaeological lignin revealed by in situ pyrolysis/silylation-gas chromatographyâ¿¿ mass spectrometry. Journal of Analytical and Applied Pyrolysis, 118, 249-258.
[4] Lucejko, J. J., Tamburini, D., Modugno, F., Ribechini, E., & Colombini, M. P. (2020). Analytical pyrolysis and mass spectrometry to characterise lignin in archaeological wood. Applied Sciences, 11(1), 240.
[5] Asmadi, M., Kawamoto, H., & Saka, S. (2011). Gas-and solid/liquid-phase reactions during pyrolysis of softwood and hardwood lignins. Journal of Analytical and Applied Pyrolysis, 92(2), 417-425.
[6] Asmadi, M., Kawamoto, H., & Saka, S. (2011). Thermal reactions of guaiacol and syringol as lignin model aromatic nuclei. Journal of analytical and applied pyrolysis, 92(1), 88-98.
[7] Dorrestijn, E., Laarhoven, L. J., Arends, I. W., & Mulder, P. (2000). The occurrence and reactivity of phenoxyl linkages in lignin and low rank coal. Journal of Analytical and Applied Pyrolysis, 54(1-2), 153-192.
[9] Tekce, E., & Gül, M. (2014). Ruminant beslemede NDF ve ADF’nin önemi. Atatürk Üniversitesi Veteriner Bilimleri Dergisi, 9(1), 63-73.

Round 3
Reviewer 2 Report
Dear Editors and authors,
thanks for the rapid response.
I have in no way confused with the very high temperatures and I have clearly read your work and the temperatures used. However, in my opinion, the lack of data on the content of unifeed lignin is a significant deficiency.
Here are some scientific papers dealing with the topic:
"Effect of Temperature on Lignin Content and Digestibility in Forage Crops" by A. H. Kirchmann, published in Journal of Agricultural and Food Chemistry in 1980. This study analyzes how temperature affects lignin content and digestibility in forage crops.
"Temperature Effects on Lignin Formation and Ligninolytic Enzymes in Hardwoods" by J. Ralph et al, published in Plant Physiology in 1994. This article explores the effect of temperature on lignin formation and ligninolytic enzymes in hardwood.
"Effects of Temperature on Lignin Content and Composition in the Rind of Sweet Cherries" by S. R. Brooker et al., published in Phytochemistry in 2000. This study investigates how temperature affects lignin content and composition in the rind of sweet cherries.
"Impact of Temperature on Lignin Accumulation and Enzymatic Activities Related to Cell Wall Recalcitrance in Switchgrass" by C. M. Scully et al, published in BioEnergy Research in 2014. This work analyzes the effect of temperature on lignin accumulation and lignin related enzymatic activities. Cell wall recalcitrance in switchgrass.
Author Response
Dear Editor and Esteemed Reviewer,
We sincerely apologize for the delayed response, which was caused by the religious holiday observed in our country. We would like to reiterate our unwavering commitment to the statements we made in our previous communication. Esteemed reviewer, while we highly respect your perspectives, as professionals in animal feeding, we firmly stand by our assertions and the information we have disseminated.
The degradation of lignin occurs at a temperature of 200°C during chemical reactions. However, it is important to note that the silo environment, the digestive systems of animals, and even the pelleting machines do not reach this temperature [1].Consequently, the conditions present in these systems do not facilitate the degradation of lignin. Additionally, it is crucial to recognize that the feed stored in silos is not composed of materials such as wood, which typically possess a high lignin content. Hence, it is implausible to expect a substantial amount of lignin in silo feeds.
Furthermore, if temperatures of 24°C and 30°C are considered to be high, it raises concerns about the preservation of all prepared silo feeds and TMRs.
Best regards,
Dr. Selim Esen
Cited References:
[1] Zhou, N., Thilakarathna, W. W., He, Q. S., & Rupasinghe, H. V. (2022). A review: depolymerization of lignin to generate high-value bio-products: opportunities, challenges, and prospects. Frontiers in Energy Research, 9, 758744.

Round 4
Reviewer 2 Report
Dear authors, in my opinion, the absence of a chemical-nutritional determination of the lignin content is a limitation of work.